# Lung Injury in COVID-19 Has Pulmonary Edema as an Important Component and Treatment with Furosemide and Negative Fluid Balance (NEGBAL) Decreases Mortality

**DOI:** 10.3390/jcm12041542

**Published:** 2023-02-15

**Authors:** Jose L. Francisco Santos, Patricio Zanardi, Veronica Alo, Vanina Dos Santos, Leonardo Bovone, Marcelo Rodriguez, Federico Magdaleno, Virginia De Langhe, Andrea Villoldo, Romina Martinez Souvielle, Julieta Alconcher, Diego Quiros, Claudio Milicchio, Eduardo Garcia Saiz

**Affiliations:** 1Intensive Care Unit, Clínica Colón, Mar del Plata, Buenos Aires 7600, Argentina; 2Cardiology Service, Clínica Colón, Mar del Plata, Buenos Aires 7600, Argentina; 3Diagnostic Imaging Service, Hospital Privado del Sur. Bahía Blanca, Buenos Aires 8000, Argentina; 4Diagnostic Imaging Service, Clínica Colón, Mar del Plata, Buenos Aires 7600, Argentina

**Keywords:** COVID-19, furosemide, edema, volume overload, diuretic, NEGBAL

## Abstract

The SARS-CoV2 promotes dysregulation of Renin–Angiotensin–Aldosterone. The result is excessive retention of water, producing a state of noxious hypervolemia. Consequently, in COVID-19 injury lung is pulmonary edema. Our report is a case–control study, retrospective. We included 116 patients with moderate–severe COVID-19 lung injury. A total of 58 patients received standard care (Control group). A total of 58 patients received a standard treatment with a more negative fluid balance (NEGBAL group), consisting of hydric restriction and diuretics. Analyzing the mortality of the population studied, it was observed that the NEGBAL group had lower mortality than the Control group, *p* = 0.001. Compared with Controls, the NEGBAL group had significantly fewer days of hospital stay (*p* < 0.001), fewer days of ICU stay (*p* < 0.001), and fewer days of IMV (*p* < 0.001). The regressive analysis between PaO_2_/FiO_2_BAL and NEGBAL demonstrated correlation (*p* = 0.04). Compared with Controls, the NEGBAL group showed significant progressive improvement in PaO_2_/FiO_2_ (*p* < 0.001), CT score (*p* < 0.001). The multivariate model, the vaccination variables, and linear trends resulted in *p* = 0.671 and quadratic trends *p* = 0.723, whilst the accumulated fluid balance is *p* < 0.001. Although the study has limitations, the promising results encourage more research on this different therapeutic approach, since in our research it decreases mortality.

## 1. Introduction

Since the beginning of the COVID-19 pandemic, the inflammatory pattern, generated by a cytokine storm [1,2,3,4] and a “COVID viral pneumonia” [5,6] has been imposed as pathophysiology of lung injury [7,8].

The COVID-19 cytokine/pneumonia hypothesis is undoubtedly fine and elegant, but it fails to explain such recurrent findings as sinus bradycardia [9], anemia [10], normotension [11], afebrile course [12,13] and increased vascular diameters [14].

Moreover, based on this approach, inflammatory-pneumonia viral, antivirals [15,16], convalescent plasma [17], tocilizumab [18,19], IL-1 receptor antagonists [20], monoclonal antibodies [21,22,23,24], were rehearsed, but only exhibit failures [12,15,16,17,18,19,20,21,22,23,24,25,26].

Thus, the pathophysiology of COVID is far from being clarified [25,27,28,29].

In addition to this, there is wide evidence that the second phase of COVID-19 would be the product of progressive hypervolemia/pulmonary edema [30,31,32,33] and not an aggravation of “pneumonia”.

Based on the foregoing, we challenged the cytokines/pneumonia paradigm.

We proposed to distinguish between what the lung injury COVID-19 appears to be -pneumonia, from what it is—pulmonary edema [33].

The hypervolemia/pulmonary edema hypothesis has a simple biological plausibility: dysregulation of the RAAS. SARS CoV 2 blocks ACE2 [34,35], induces its downregulation [36,37], generates an imbalance in the RAAS [38,39,40,41] with a harmful increase in angiotensin II [42,43] and a decrease in angiotensin 1–7 [44,45] and a harmful increase in aldosterone [46]. The result is a decrease in urine output [40] and excessive retention of sodium and water, producing a state of noxious hypervolemia with subsequent pulmonary edema [30,31].

In this setting of detrimental overload, a different approach emerged: lung injury in COVID-19 is a pulmonary edema with a “dual phase” [31,47]. A “first phase” of viral pneumonitis [25]—increase of the alveolar capillary membrane permeability. However, the “second phase” is most harmful and stealthy: high pressure pulmonary edema, caused by increase of hydrostatic pressure [48,49,50] secondary to volume overload phase 2, secondary to dysregulation of the RAAS [34,51]. This results in a “dual phase” that triggers acute pulmonary edema. If this pulmonary edema is not resolved, then a “third phase” comes, with the typical ARDS [52].

A new hypothesis that could explain COVID-19 lung injury must rely on evidence. Such evidences are:

Pathological anatomy: there are multiple descriptions of severe capillary congestion and alveolar flooding [47,53,54,55].

Clinical: in a second phase, many patients evolve afebrile [12,13], normocardic and, normotensive [4,11,33]. We would be facing a tempest without thunderbolts or lightning.

There are multiple hypotheses for sinus bradycardia in COVID-19 [9,56] but one explanation is enough: with stable cardiac output and in the presence of hypervolemia, systolic volume increases and, according to Frank Starling’s Law, there is a compensatory decrease in the heart rate.

Biochemical: Anemia is dilutional and the increased hypervolemia leads to more anemia and therefore worse prognosis [10,57]. Something similar occurs with lymphopenia [58,59,60], which would have a dilutional component. 

Tomographic: in chest tomography (CT) hypervolemia is confirmed by dilation of the great vessels [33,61], of the cardiac axis, without heart failure [33,61], and of the pulmonary vessels [14]. In addition, the rapid improvement of the CT Score with negative fluid balance (NEGBAL) is strong evidence that there is more pulmonary edema than pneumonia [33].

Cardiotechnics: dilation of the right ventricle [62,63] and increased end-diastolic volume [31] are clear manifestations of hypervolemia.

Techniques thermodilution: the increase in the rate of extravascular lung water (EVLW) in COVID-19 has already been demonstrated [30,31].

Therapeutic: improved oxygenation, decreased cardiac axis and great vessels, and increased hematocrit are results obtained with negative balance [33].

Despite this evidence, the role of hypervolemia and edema in lung injury in COVID-19 continues to be underestimated in daily practice. Taking into consideration what was previously stated, we detected pulmonary edema and, consequently, in the presence of pulmonary edema in COVID-19 patients, we established a standard treatment, consisting of oral hydric restriction and diuretics [48,64,65] called NEGBAL (negative fluid balance). 

In this research, we compared the evolution of patients with moderate and severe COVID-19 infection into two groups, a control group and a group that received the NEGBAL approach.

At the time of submission of our study, excepting our previous report [33], we did not find any literature that proposed the model of volume overload pulmonary edema secondary to the dysregulation of the RAAS in COVID-19.

## 2. Materials and Methods

### 2.1. Ethical Approval

The study was conducted according to the guidelines of the Declaration of Helsinki and its later amendments, and approved by the Ethics in Investigation Committee of INSTITUTO DE INVESTIGACIONES CLINICAS, Mar del Plata, Argentina (OHRP: IRB 00008222, date of approval: 3 September 2021).

### 2.2. Study Design and Setting

This single-center retrospective, observational case–control design study took place during the period of 8 July 2020 to 15 February 2022. One-to-one matching was included, all with confirmed diagnosis of COVID-19, admitted to our high complexity center in Mar del Plata, Argentina. 

A one-to-one case control ratio was designed.

The cases were all patients who had received the NEGBAL approach since we started implementing it, 24 June 2021 to 15 February 2022. 

Control patients were selected among those who did not receive the NEGBAL approach, since they were admitted to the Intensive Care Unit before 24 June 2021, when the NEGBAL strategy was implemented. 

### 2.3. Data Compilation

We reviewed data from medical records: demographic; clinical, laboratory; blood gas; chest tomography (CT); the oxygen therapy support; and mechanical ventilation (IMV) requirements, all of which were reviewed and recorded by investigators. Clinical outcomes were followed up to 1 May 2022. 

In this case–control study, we compared fluid management applied during eight days in patients with acute lung injury for COVID-19: a conservative strategy for the control group, and NEGBAL approach for the NEGBAL group.

The control group received standard treatment. It was based on dexamethasone 6 mg/day, enoxaparin 40 mg/day, oxygen supplement (nasal cannula, reservoir mask, HFNC), invasive or non-invasive mechanical ventilation and standard care. 

The NEGBAL group underwent, in addition to the standard care, a treatment with furosemide in continuous intravenous infusion: the NEGBAL approach.

Inclusion criteria were as follows: (1) age older than 18 years; (2) confirmed diagnosis of COVID-19 through real-time reverse transcriptase polymerase chain reaction (RT-PCR) assay with samples obtained from nasopharyngeal swab or positive antinucleocapsid IgM antibodies; (3) PaO_2_/FiO_2_ (ratio of arterial oxygen partial pressure to fractional inspired oxygen) < 300; and (4) tomographic evidence of acute pulmonary edema, defined as ground glass infiltration, dilated superior vena cava, large pulmonary arteries, and dilated right ventricle or dilated cardiac axis.

Exclusion criteria were as follows: (1) patients with indication for diuretics for another reason; (2) renal failure; (3) cardiac failure; (4) hepatic failure; (5) hypernatremia or hyponatremia; (6) hypotension, use of inotropics or shock; (7) receiving antivirals or monoclonal antibodies.

CTs were performed upon admission to ICU, CT controls were scheduled for day 4 (+/− 1 day), day 8 (+/− 1 day), and day 12 (+/− 1 day). A 1.5 mm slice thickness and 1.5 mm interval were used for the axial image.

For the evaluation of CT infiltrates, the scoring was as described by Pan F et al. [66]. The measurement of the superior vena cava diameter (Ø svc) in the CT was performed just above the arch of the azygos veins. 

### 2.4. Negative Fluid Balance (NEGBAL) Approach

We established a treatment NEGBAL approach: It consisted of oral hydric restriction and diuretics (20 mg of furosemide, intravenous bolus, followed by furosemide in endovenous continuous infusion, starting at 60 mg/day). The objective was to achieve a negative fluid balance, between 1000 to 1500 mL/day adjusted to body surface area, with a final target of 10% of body weight in 8 days. The presence of hypotension (systolic blood pressure less than 100 mmHg for 30 min), use of inotropics or elevated serum creatinine higher to 2.0 mg/dl was considered a cause of suspension of NEGBAL. All patients were followed until either death or complete recovery and discharge were reached.

The primary clinical outcome was mortality within 28 days after ICU admission. Secondary outcomes were days of invasive mechanical ventilation; days of ICU stay, and days of hospital stay, to measure accumulated fluid balance at day 8 (AccFluBal), PaO_2_/FiO_2_, CT Score, diameter SVC, hematocrit and lymphocytes evolution. 

Safety outcomes included treatment-emergent adverse events, serious adverse events, and discontinuations of NEGBAL approach.

### 2.5. Statistical Analysis Plan

#### 2.5.1. Data Source and Description

We analyzed data from 116 patients treated for COVID-19 disease. A first group of 58 patients was treated with the standard treatment (CONTROL group) and a second group was treated following the NEGBAL approach (NEGBAL group). At this stage, we compared data from both populations as to inquire whether there was any statistically significant difference between the evolution of those patients treated with NEGBAL protocol and those treated with the standard treatment.

#### 2.5.2. Baseline Data

Descriptive data are presented as means (std) or medians (with range) for continuous data depending on the distribution of the variable. Categorical variables are presented as frequencies and percentages.

#### 2.5.3. Primary Analysis

The variables analyzed were: age, weight, sex, vaccination status for 1st and 2nd doses obesity diabetes instead COPD or tabaquism, cardiopathy instead, HR (heart rate), SBP (systolic blood pressure), DBP (diastolic blood pressure), temperature instead, RR (respiratory rate), APACHE (acute physiology and chronic health evaluation score), BMI (body mass index), SSD (Symptom Start Date), PO_2_, FiO_2_, PaFiO_2_, PCO_2_, HCO_3_, creatinine instead lymphocytes instead, plasmatic sodium instead, hematocrit instead, arterial hypertension, BNP (pro-B-type Natriuretic Peptides), troponin instead TGO (glutamic oxaloacetic transaminase), TGP (glutamic pyruvic transaminase), ALP (alkaline phosphatase), total bilirubin instead, direct bilirubin instead.

The references NEGBAL and CONTROL refer to which group the variable under consideration belongs and ADM means that the value of the variable was observed at the moment of admission.

The data consisted of: PaFiO_2_, creatinine and hematocrit measurements for at the moment of admission, at days 4 and 8 and at discharge; leukocytes at the moment of admission; accumulated hydric balance after the 8 days; hospitalization days, ICU days, days under IMV and IMV (invasive mechanical ventilation) use; and result of the treatment (RES, alive or dead); CT score and superior vena cava diameter (CT SCORE and SVC, respectively) measurements previous to the treatment, at the moment of admission, at days 4, 8 and 12.

The comparison between numerical variables was made via Welch’s two samples *t*-test with null hypothesis that the difference of the means equals 0. The comparison between categorical variables was carried out mostly via a Person’s χ2 test, except for the case of the CARDIOP variable, for which a Fisher’s test was used.

Five types of analyses were performed:
An intrinsic analysis via paired differences *t*-tests regarding the evolution of the variables PaFiO_2_, creatinine and hematocrit at the moment of admission, days 4 and 8 and discharge and CT SCORE and VSC (superior vena cava) previous to the treatment, at the moment of admission, and at days 4, 8 and 12 for the same group;A comparative analysis via unpaired differences *t*-tests regarding the same variables as above (as well as the variable leukocytes at the moment of admission) but comparing for each day the values of both groups, together with a comparison (again using unpaired difference *t*-tests) of the variables accumulated hydric balance (ACC HYD BAL) and hospitalization days between groups and a Pearson’s χ2 test for the result;A simple regression model to assess the correlation between PaFiO_2_ and ACC HYD BAL in both groups.A multivariate regression model to assess the correlation between PaFiO_2_ and ACC HYD BAL and vaccination (VAC) for the whole population.Paired differences *t*-tests comparing the values of the variable ACC HYD BAL for survivors and non survivors within each group and for the whole population.


#### 2.5.4. Results

For each numerical variable, two tables and one plot are provided: the corresponding statistical summary (IQR standing for interquartile range and sd for standard deviation) for the Control and BLANEG groups and the *t*-test for difference of the means and the boxplot for both samples. For the categorical variables, a contingency table is provided, and on its caption the result of the χ2 test (Fisher in the case of CARDIOP).

The results are divided in subsections according to the analyses detailed in the previous paragraph: intrinsic evolution, day by day comparison and regression.

All analyses were performed using R software, version 4.1.1 (R Foundation for Statistical Computing).

(Please see Appendix A: Statistics Nº 1 and Nº 2).

## 3. Results

### 3.1. Population Analysis

A total of 293 patients with COVID-19 were referred to our ICU. After excluding 89 patients with PaO_2_/FiO_2_ > 300, 27 for prior indication of diuretics, 6 with shock, 9 with limitation of therapeutic effort, and 46 with hepatic, kidney or heart failure; we included in the final analysis 116 patients withCOVID-19 lung injury. There were not any dropouts in the study population.

Twenty eight patients died during hospitalization and 88 were discharged. Mean (SD) age was 59.7 (±13.6) years, ranging from 22 to 91. Forty were female (34.5%) *p* = 0.55. Among the reported population, 92 (79.3%) had comorbidities; the most common ones were obesity (50.9%) and hypertension (40.5%), followed by diabetes mellitus (25.9%) and COPD (17.4%). 

Fifty two patients had received the first dose of vaccine (44.8%) and 27, (23.2%) the second dose. The mean (SD) BMI was 30.1 (5.16). The mean (SD) hematocrit was 39.8% (4.91%). The mean (SD) PaO_2_/FiO_2_ on admission was 164 (70) and the mean (SD) CT score was 15.7 (4.63).

There were no significant differences in demographics, clinical, tomographic biochemical and background variables between the cases and the control group at the time of ICU admission, except for the prevalence of vaccinated patients, which was higher in the NEGBAL group (*p* = 0.01). (See demographic information and biochemical data at admission information of both groups in Table 1).

The percentage of the patients with moderate or severe lung injury at admission were:

In the Control group: 33% (n = 19) of moderate lung injury and 67% (n = 39) of severe lung injury. In the NEGBAL group: 31% (n = 18) of moderate lung injury and 69% (n = 40) of severe lung injury.

The causes of death in the population studied were:

Control group: 22 deaths: 5 secondary to refractory distress, 6 due to sepsis, 8 due to refractory distress and sepsis and 3 due to multiple organ failure. 

NEGBAL Group: 6 deaths: 4 secondary to sepsis, 1 due to coronary disease and 1 due to ventilator-associated pneumonia.

No patient in the NEGBAL group presented criteria for suspension of the NEGBAL protocol. In other words, no patient had hypotension, no patient had tachycardia, no patient had renal failure, nor sodium disorders.

For the evolutionary analysis, we compared day of admission, day 4, day 8 and day of discharge.

### 3.2. Primary Outcome

Analyzing mortality: it is observed that the NEGBAL group has lower mortality than the control group, *p* = 0.001. See Figure 1A.

### 3.3. Secondary Outcomes

Compared with controls, NEGBAL has significantly fewer days of hospital stay, fewer days of ICU stay, and fewer days of IMV (see Figure 1B–D)

Comparing accumulated fluid balance (AccFluBal) at day 8: Both groups had negative fluid balance; however, the NEGBAL group had significantly higher negative AccFluBal −8027 ± 2516 mL versus AccFluBal Control, −685 ± 5045 mL (see Figure 2A).

Comparing within the control group, the AccFluBal at day 8 between survivors and non-survivors. We observed that survivors in the control group developed negative fluid balance, (spontaneous, without diuretics -”auto-NEGBAL”-) that was significantly higher −3565 ± 1000 mL than non-survivors: +580 ± 1000 mL, with *p* < 0.001. (see Figure 2B).

The linear regression analysis of the NEGBAL group showed a statistically significant correlation between PaO_2_/FiO_2_ and accumulated fluid balance (AccFluBal) *p* = 0.04 (see Figure 2C)

The linear regression analysis of the control group also showed a statistically significant correlation between PaO_2_/FiO_2_ and accumulated fluid balance (AccFluBal) *p* = 0.007 (see Figure 2D)

Regarding the multivariate model, the *p*−value corresponding to the whole model was *p* < 0.001. The *p*−values for the variables AccFluBal and vaccination (Vac) account for whether or not such variables are significant explaining the variation of the variable PaO_2_/FiO_2_. The results obtained for the variable AccFluBal was *p* < 0.001. As for the variable Vac, the *p* values were 0.67 for the linear and 0.72 for the quadratic fit, respectively.

### 3.4. Complementary Analysis

We analyzed the temporal evolution of PaO_2_/FiO_2_ between admission, day 4 and 8. In the NEGBAL group, PaO_2_/FiO_2_ showed significant progressive improvement coinciding with the application of the NEGBAL approach, while control showed no evolutionary differences in PaO_2_/FiO_2_ (Figure 3A,B).

We compared PaO_2_/FiO_2_ between NEGBAL and control. Significant differences were detected with better oxygenation in NEGBAL at days 4 and 8. (Figure 3A,B).

We analyzed the temporal evolution of the CT score between admission, days 4 and 8. In NEGBAL, the CT score showed a significant progressive decrease, coinciding with the application of the NEGBAL approach. Meanwhile, the control group did not show significant evolutionary differences in the CT score. (Figure 3C,D).

We compared the CT score between NEGBAL and control between admission, day 4 and 8. Significant differences were detected with a greater decrease in the CT score in the NEGBAL group. (Figure 3C,D).

We analyzed the temporal evolution of the diameter of the SVC (Ø SVC) and hematocrit (HCT) between admission, day 4 and 8 can be seen in Figure 3E–H, respectively. 

### 3.5. Analysis Variable Security

We found no differences in safety measurements between both groups.

No patient in the NEGBAL group presented criteria for suspension of the approach.

More detailed results analysis can be found at Table 2.

## 4. Discussion

Our study has some limitations. First, due to the study design, it is retrospective and one-center only. Second, the interpretation of our findings might be limited by the sample size. Third, a weakness of this study is not being able to report the variants of SARS-CoV2 that affected each of the patients. Unfortunately, viral typing is out of the investigators reach and the means we currently have.

Unlike high-altitude pulmonary edema (HAPE), oxygen administration alone may be ineffective in severe cases of COVID-19 due to extensive alveolar cell compromise that reduces the gas exchange surface area. It is also important to note that alveolar destruction by pneumolysis induced by initial intraalveolar viral replication hinders spontaneous resorption of pulmonary edema.

The focus on the link between SARS-CoV2 and RAAS imbalance was mentioned by Rothlin et al. The use of ACEI/ARBs is promoted to “block” the ACE2 receptor, and decrease the proinflammatory activity of angiotensin II, although ACEI/ARBs would have a protective effect and would not be useful in advanced conditions [29].

The NEGBAL group comparison to the control group describes homogeneous groups without significant differences, except for the prevalence of vaccinated people. This was due to the fact that the advance of vaccination in our country coincided with the application of the NEGBAL approach. However, this did not affect the results; first, because the objective of the vaccines is not the improvement of lung injury, but its prevention [67,68]; second, multivariate analysis did not show that vaccination reverses lung injury.

Our population upon admission to the ICU presented itself without clinical evidence of cytokine storm. However, our study does describe several clinical, biochemical and tomographic findings consistent with the model hypervolemia/pulmonary edema:

The control group showed that there is a tendency to maintain the Ø SVC or even a progressive increase, not significant, of the Ø SVC—evidence of hypervolemia—and associated with progressive drop in PaO_2_/FiO_2_ and an increase in the CT score—evidence of pulmonary edema.

On the contrary, in the NEGBAL group, fluid overload was reversed, through negative balance, observing a significant progressive decrease in the Ø SVC—evidence of resolution of hypervolemia—and progressive improvement of PaO_2_/FiO_2,_ which is positively correlated in the linear regression analysis with the magnitude of the negative balance.

CT score also improved—evidence of resolution of the pulmonary edema—which, as an expected consequence, resulted in a significant decrease in IMV requirements in the NEGBAL group. 

Signs of hypervolemia are also described by Lang et al. [14], Eslami et al. [61], and Argulian and Yu described right ventricle dilation [62,63]. Although the deregulation of the RAAS in COVID-19 is out of the question, its role in the genesis of pulmonary injury is underestimated, despite the fact that the relationship between hypervolemia and pulmonary edema in COVID-19 is described by Villard et al. [46] when she demonstrated that “*elevated levels of aldosterone (excessive fluid retention) in COVID-19 are associated with greater pulmonary injury (pulmonary edema)*”. Even Rysz [40] described the genesis of hypervolemia in pigs, generated by ACE 2 deficiency and high levels of angiotensin II, measuring water retention and detecting a decrease of eighty percent in urine output. 

The control group showed a decrease in the hematocrit and lymphocytes level, also coinciding with an increase Ø SVC. These three simultaneous observations can only be explained in a simple way by the presence of hypervolemia that generates anemia and lymphopenia by a dilutional phenomenon. On the contrary, in the NEGBAL group, as the negative balance progressed, the hypervolemia was corrected, and the desired result appeared: Ø SVC decreased and hematocrit and lymphocytes increased [33,69].

On admission, the study population had normal heart rate or sinus bradycardia associated with decreased hematocrit and increased Ø SVC. This association has no simpler explanation than the presence of hypervolemia. Consequently, sinus bradycardia in COVID-19 is the result of Frank Starling’s Law [70]. In fact, Rhi Shi [31] observed that COVID-19 patients had a higher end-diastolic volume, but lower lactate than non-COVID-19 patients. This demonstrated hypervolemia, not hemodynamic failure.

This early improvement in the CT score and PaO_2_/FiO_2_ with NEGBAL can be attributed to resolution of the edema. Furthermore, it is unusual for a “pneumonia” to improve tomographically on day 4. Sibbald et al. [48] described the relationship between hypervolemia and pulmonary edema. Similarly, Rasch et al. [30] reported *“that COVID-19 lung injury has up to five times more EVLW index -2600 mL- than normal lungs -500 mL”*-. Rhi Shi et al. [31] reported the same facts. 

We noted that within the control group, the surviving subgroup had a higher spontaneous negative fluid balance during hospitalization (without diuretics, “auto-NEGBAL”), compared to the non-surviving control subgroup, which had a positive fluid balance. This “auto-NEGBAL” could result from the natural regeneration of ACE2 [71], restoring the RAAS and euvolemia, naturally.

The NEGBAL group quickly and effectively resolved the pattern of hypervolemia/pulmonary edema in COVID-19 and this was remarkable when comparing the hospital evolution in both groups: the NEGBAL group has fewer days of MV, shorter ICU stay and shorter hospital stay than the control group.

Finally, we are convinced that the resolution of the hypervolemia/pulmonary edema pattern through the NEGBAL approach is the reason why the analysis of the primary outcome showed that COVID-19 patients in the NEGBAL group have lower mortality than the control group and less use of IMV, confirmed with the multivariate analysis showing the relationship between better PaO_2_/FiO_2_ and negative fluid balance. This agrees with many authors describing the benefit of negative fluid balance in lung injury of any kind [64,65,69,72,73,74].

## 5. Conclusions

Our study has limitations. In any case, the hypervolemia/pulmonary edema hypothesis has the advantage that it is easy to objectify, and it explains and resolves signs and observations associated with COVID-19 in a simple way. The negative fluid balance in our study showed an improvement in oxygenation, shorter hospital stay, less use of IMV and lower mortality, with the advantage of being an inexpensive and accessible therapy. We invite you to challenge this hypothesis and its therapeutic line, instead of being passive subjects.

## Figures and Tables

**Figure 1 jcm-12-01542-f001:**
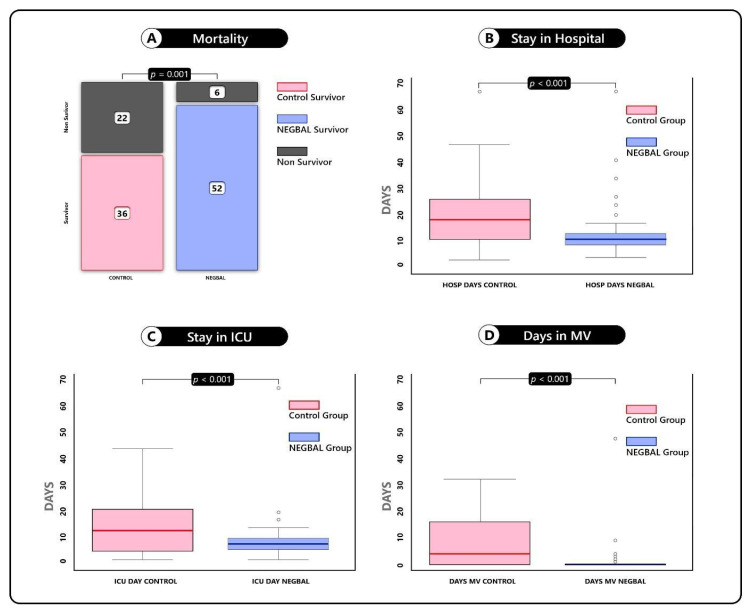
Box plot of comparative analysis of mortality and length of stay. (**A**) Mortality. (**B**) Stay in Hospital. (**C**) Stay in ICU. (**D**) Days in IMV.

**Figure 2 jcm-12-01542-f002:**
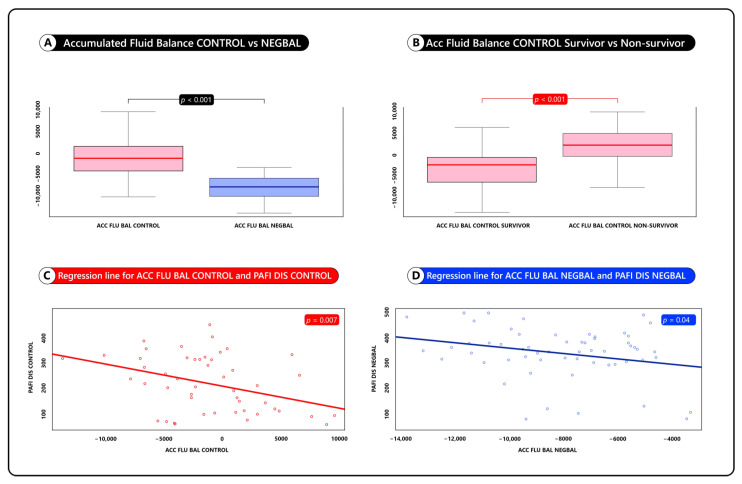
Accumulated fluid balance analysis. (**A**) Box plot of accumulated fluid balance CONTROL vs. NEGBAL. (**B**) Box plot of accumulated fluid balance CONTROL survivor vs. non-survivor. (**C**) Regression line for ACC FLU BAL CONTROL and PAFI DIS CONTROL. (**D**) Regression line for ACC FLU BAL NEGBAL and PAFI DIS NEGBAL.

**Figure 3 jcm-12-01542-f003:**
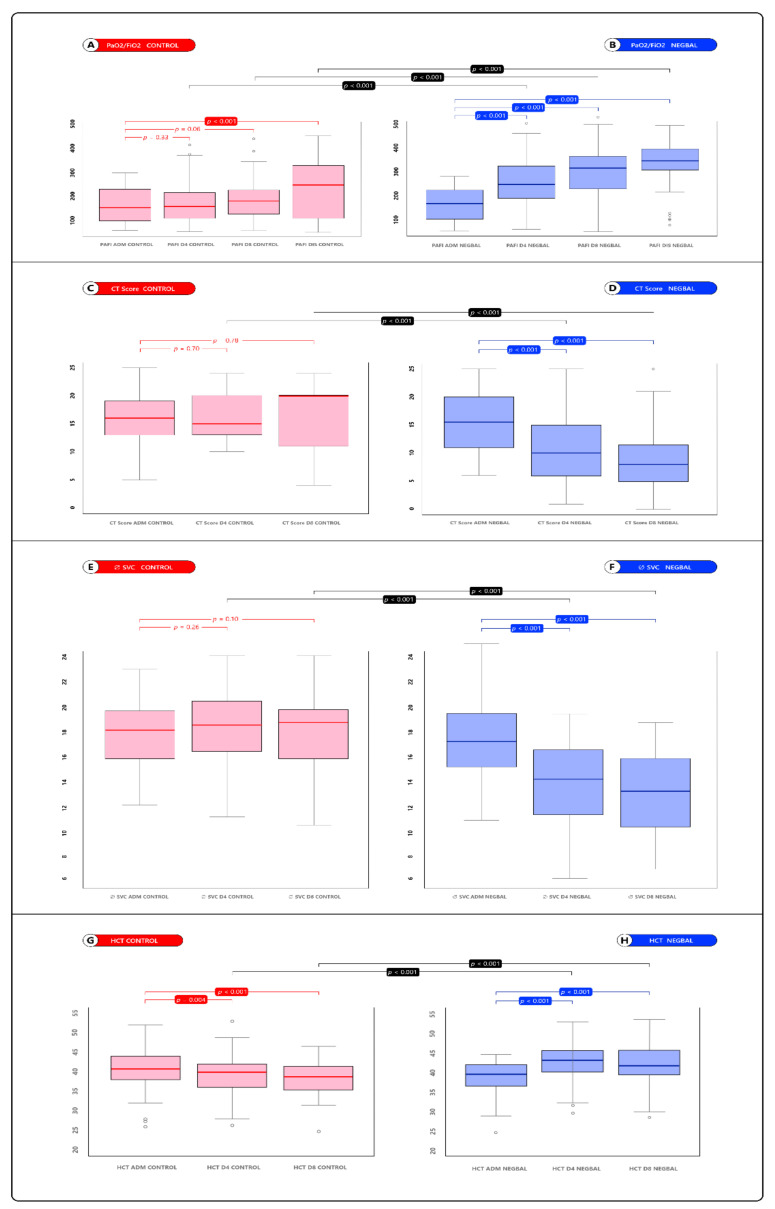
Box plot of comparative analysis of PaO_2_/FiO_2_, CT score, diameter SVC and hematocrit. *(***A**) PaO_2_/FiO_2_ CONTROL; (**B**) PaO_2_/FiO_2_ NEGBAL; (**C**) CT Score CONTROL; (**D**) CT Score NEGBAL; (**E**) Ø SVC CONTROL; (**F**) Ø SVC NEGBAL; (**G**) HCT CONTROL; (**H**) HCT NEGBAL.

**Table 1 jcm-12-01542-t001:** Comparative analysis of population with COVID-19 at ICU admission.

Variable	Total Population	CONTROL Group n = 58	NEGBAL Group n = 58	*p* Value
Demographics—Pathological antecedents Variables				
Age	59.6 ± 13.6	59.45 ± 12.04	59.86 ± 15.14 (100)	0.87 (−5.44, 4.61)
Obesity	59 (50.8%)	30 (51.7)	29	1
BMI	30.1 ± 5.16	30.25 ± 5.46	29.93 ± 4.89	0.74 (−1.58, 2.22)
AHT	47 (40.5%)	24	23	1
DBT	30 (25.8%)	17	13	0.52
COPD	20 (17.3%)	10	10	0.52
COVID-19 Day	8.41 ± 3	8.93 ± 2.7	8 ± 3.2	0.75
VAC-1	52 (44.8%)	18	34	0.01
VAC-2	27 (22.4%)	5	22	0.01
Clinical variables				
Heart Rate	83.5 ± 15	84.55 ± 14.9	82.5 ± 15.19	0.46 (−3.48, 7.58)
Systolic BP	127 ± 19.2	127.2 ± 20.77	128.3 ± 17.75	0.75 (−3.48, 7.58)
Diastolic BP	77.4 ± 12.4	76.79 ± 12.47	78.14 ± 12.4	0.56 (−5.92, 3.23)
Temperature	36.5 ± 0.86	36.63 ± 0.88	36.43 ± 0.84	0.20 (−0.11, 0.52)
Resp. Rate	24.9 ± 4.58	25.34 ± 4.61	24.47 ± 4.56	0.30 (−0.80, 2.56)
APACHE II	8.01 ± 3.26	7.6 ± 3.04	8.43 ± 3.44	0.17 (−2.02, 0.36)
Biochemical variables				
Po2	73.3.± 18.8	73.99 ± 21.55	72.8 ± 15.97	0.73 (−5.79, 8.17)
Pafio2	164 ± 70.2	164.81 ± 73.64	163.6 ± 67.26	0.92 (−24.6, 27.1)
Pco2	33.28 ± 5.15	33.75 ± 5.40	32.81 ± 4.90	0.32 (−0.95, 2.83)
HCO3	21.8 ± 3.06	21.64 ± 3.01	23.14 ± 3.12	0.38 (−1.62, 0.63)
Creatinine	9.01 ± 2.22	9.41 ± 2.38	8.75 ± 2.01	0.11 (−0.15, 1.47)
Hematocrit	39.7 ± 4.9	40.38 ± 5.22	39.20 ± 4.54	0.19 (−0.62, 2.97)
Lymphocytes	959 ± 665	963 ± 773	955 ± 540	0.42 (−345, 145)
Troponina	10.4 ± 8.35	10.84 ± 8.53	9.99 ± 8.25	0.63 (−2.73, 4.43)
proBNP	224 ± 235	277 ± 246.19	188.80 ± 222.86	0.11 (−22.4,198)
GOT	49.9 ± 37.7	53.84 ± 38.85	45.15 ± 36.39	0.26 (−6.56, 23.9)
GPT	57.2 ± 68	60.69 ± 64.60	53.38 ± 71.96	0.58 (−20.18, 35.3)
Chest Tomography Score and Diameter Superior Vena Cava variables				
CT Score	15.7 ± 4.63	15.8 ± 4.14	15.6 ± 5.08	0.80 (−1.51,1.94)
Ø SVC	17.7 ± 2.97	18.1 ± 2.55	17.3 ± 3.29	0.14 (−0.28,1.93)

Abbreviations: BMI: Body Mass Index; DBT: Diabetes; COPD: Chronic Obstructive Pulmonary Disease; COVID-19 Day: Day since the beginning of COVID-19, in which it was admitted to ICU; VAC-1: Vaccine first dose; VAC-2: Vaccine second dose; BP: Blood Pressure; HCT: Hematocrit at admission; AHT: Arterial Hypertension; LYMP: Lymphocytes; Pafio2: PaO_2_/FiO_2_ ratio; proBNP: pro-B-type Natriuretic Peptides; GOT; glutamic oxaloacetic transaminase; GPT, glutamic pyruvic transaminase; CT Score: Chest Tomography Score; Ø SVC: diameter Superior Vena Cava.

**Table 2 jcm-12-01542-t002:** Comparative analysis during evolution of COVID-19.

Variable	CONTROL Group n = 58	NEGBAL Group n = 58	*p* Value
Biochemical variables			
PaPafio2 day 4	175 ± 90.5	249 ± 100	<0.001 (−109, −37.8)
PaPafio2 day 8	183 ± 87	290 ± 101	<0.001 (−146, −67.2)
PaPafio2 day dis.	235 ± 116	344 ± 97.6	<0.001 (−141, −61)
HCT day 4	39.1 ± 5.21	43.1 ± 5.21	<0.001 (−5.90, −2)
HCT day 8	38.5 ± 4.67	42.5 ± 5.23	<0.001 (−6.38, −2.05)
LYMP day 4	697 ± 402	1056 ± 735	0.002 (−586, −130)
CREAT day dis.	13.1 ± 9.8	11.6 ± 8.2	0.37 (−1.84, 4.85)
Chest Tomography Score and Diameter Superior Vena Cava			
CT Score day 4	16.4 ± 4.31	10.6 ± 5.96	<0.001 (3.31, 6.8)
CT Score day 8	16.3 ± 5.79	8.77 ± 5.57	<0.001 (4.18, 10.9)
Ø SVC day 4	18.9 ± 3.1	13.8 ± 4.09	<0.001 (3.29, 6.8)
Ø SVC day 8	17.8 ± 3.42	12.7 ± 3.43	<0.001 (3.14, 7.21)
Accumulated Fluid Balance			
ACC FLU BAL	−685 ± 1000	−7898 ± 1000	<0.001 (5419, 8453)
Hospital stay, ICU stay and IMV days			
Hospital Stay	20.5 ± 13.2	13 ± 13.2	<0.001 (3.15, 11.8)
ICU Stay	15.7 ± 13	9 ± 8.91	<0.001 (2.77, 10.3)
Days of IMV use	4 ± 10.2	1 ± 6.4	<0.001 (4.20, 10.5)
Patients IMV use	31	9	<0.001
Mortality			
Mortality	22	6	0.001
Accumulated Fluid Balance CONTROL group			
Control Group	SURVIVORS Control group	NON SURVIVORS Control group	*p* value
ACC FLU BAL	−3685 ± 1000	+5898 ± 1000	<0.001 (−7648,−2688)

Abbreviations: SD: Standard Deviation; Pafio2: PaO_2_/FiO_2_ ratio; HCT: Hematocrit; LYMP: Lymphocytes; CREAT: creatinine; CT Score: Chest Tomography Score; Ø SVC: diameter Superior Vena Cava; ACC FLU BAL: Accumulated Fluid Balance at day 8; IMV: Invasive Mechanical Ventilation.

## Data Availability

The data presented in this study are available on request from the corresponding author. More data is contained within the Appendix A. The visualization of the tomographies is available in an online website.

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
