# Peer review of "Lung Injury in COVID-19 Has Pulmonary Edema as an Important Component and Treatment with Furosemide and Negative Fluid Balance (NEGBAL) Decreases Mortality"

_jcm, 2023, doi:10.3390/jcm12041542_

Round 1

Reviewer 1 Report

In this study Santos et al. affirm that lung injury in covid19 disease is lung edema and negative fluid balance improves outcome. They studied two groups of patients, one receiving standard care (control group), the other continuous infusion of furosemide (NEGBAL group), and compare outcome. The supposition is that covid19 causes dysregulation in the RAAS leading to fluid overload. The authors propose negative fluid balance as a treatment to resolve pulmonary edema.

The hypothesis is challenging but the design of the study has one big limitation. Due to the retrospective nature of the study, control patients were enrolled between July 2020 and June 2021, while NEGBAL patients between June 2021 and February 2022. This temporal division is misleading. The sars-cov2 virus evolves during pandemic and the disease transformed: it is not only about vaccine, but new less severe variants of the virus spread, the therapeutical and mechanical ventilation approach changed, the mortality dramatically decreased. Indeed, the authors observed a decrease in mortality in the NEGBAL group, but all the above-mentioned factors should be considered as possibly confounders (especially new variants of virus). Without adjunctive data characterizing population (see major comments) the two groups are not comparable.

Concerning the results, the early start of furosemide seems to be promising in this population, but no assumptions can be made about the type of lung injury below. Lung injury could be ARDS, as reported by most of the literature, and negative balance work as well.

Finally, the development of complications and the cause of death in the two groups were not mentioned, it could be interesting to know if patients dye for lung injury or other comorbidities.

Major comments

1)      The title: “lung injury in COVID-19 is Pulmonary Edema” is a strong statement. Covid-19 is a complex systemic disease which involves lungs with development of an ARDS-like injury characterized by pulmonary edema, inflammation, microvascular abnormalities, etc. I suggest changing the title saying that lung edema is one of the main features in covid19 lung injury.

2)      Line 70-76: The hypothesys that lung injury is caused by pulmonary edema secondary to RAAS dysregulation is very interesting and studied by many authors. However, the use of ACEI/ARBs in covid19 patients is controversial, with some authors promoting the utilization of ACEI/ARBs to “block” the ACE2 receptor, and other encouraging discontinue of these drugs in chronic patients because of an overexpression of the receptor. This should be discussed. In this scenario the supposed pathophysiology of the “fluid overload” remains speculative, lacking evidence about alteration in RAAS system, i.e. measured levels of RAAS mediators, measured urine output before enrollment. Without supporting data, the assumptions are very weak.

3)      Line 110-114: the selection of the patients is not clear. You exclude your population has already developed ARDS based on what? The inclusion criteria of your study meet the Berlin definition of ARDS as well (P/F<300, chest imaging, temporal evolution, not cardiogenic edema). About this, how many patients were ventilated in the two groups? How many patients required PEEP, and which level? How many days after onset of symptoms were patients enrolled? These data are missing.

4)      Line 129-138: The study design is weak: without further informations about population the two groups are not comparable. What were virus variants responsible for infections in the two groups? Do you have typing of the variants? How many patients were ventilated in each group at admission? What were was the percentage of the patients with mild moderate or severe lung injury at admission? How many patients required mechanical ventilation during ICU stay?

5)      Line 157-158: you define “fluid overload” based on diameter of superior vena cava, please provide references about cut-off. The normal diameter of SVC is up to 22 mm, and you have lower values. Moreover, you lack dynamic indices of fluid overload (i.e. SVC collapsibility) or hemodynamic monitoring (i.e. CVP). Do you have this information?

6)      Line 232-233: provide numerosity at each timepoint? How many dropouts?

7)      Line 235-236: cite the cause of death and development of complications in each group.

8)      Line 265-266: how many patients on MV for each group?

9)      Line 267-267: -8 L in the NEGBAL group is a very huge negative balance. It remains speculative how such a negative fluid balance improved outcome, moreover development of complications was not mentioned. If the hypothesis is “fluid overload” and the diuretic therapy starts at admission, the patients should have accumulated fluids before admission: how many days after onset of symptoms were the patients enrolled? did they present any other sign of fluid overload (i.e.: peripheral edema)? The significant reduction of SVC in this group is expected, but it doesn’t mean that there was fluid overload. Also in this case, some hemodynamic data are missing. Moreover mechanical ventilation settings may influence the possibility to target negative balance (if patients require high PEEP), it is crucial to define ventilatory support in the population.

10)  Line 360-362: discuss better the limitations of the study

11)  Line 373: the progressive increase of SVC diameter and the CT score in control group are not significant, you can’t discuss this.

12)  Line 378: “progressive reduction of hypovolemia”, provide evidence of hypervolemia

13)  Line 408-409: it may depend on the severity of the pathology.

Minor comments:

1)      Line 40: you missed the subject “Analyzing mortality, we observed that”

2)      Line 83-84: I would define ARDS as an evolution of the pathology more than a “third hit”

3)      What does EPOC mean, define abbreviations in the tables

4)      Specify how you presented data in the figure legends (box-plot)

Reviewer 2 Report

The article entitled "Lung Injury in COVID-19 is Pulmonary Edema and treatment 2 with Furosemide and Negative Fluid Balance (NEGBAL) de- 3 creases mortality" postulates a novel therapeutic strategy based on the interpretation of the pulmonary compromise as pulmonary edema associated with vascular and renal alterations in severe COVID-19. The statistical results of lower mortality certainly speak of a favorable outcome with the proposed fluid balance management. However, the pathophysiology of severe COVID-19 hypoxemia must be explained further, strengthening the hypothesis. Pulmonary edema in COVID-19 has been shown to have similarities with High Altitude Pulmonary Edema, which results from acute ascent to high altitude. In the case of HAPE, diuretics are not required as the lung tissue is practically fully functional, and a reduction of the pulmonary pressure achieved through oxygen administration is very effective in leading to the edema's reabsorption. Nevertheless, it is important to express that reducing the increased pulmonary artery pressure only through the administration of oxygen can be ineffective in severe cases of COVID-19 due to the extensive alveolar cell compromise reducing the gas exchange surface area. It is also important to stress that alveolar destruction due to pneumolysis induced by the viral intra-alveolar replication makes reabsorption of the pulmonary edema very difficult, if not impossible, as filtration continues throughout the disease.

https://link.springer.com/article/10.1007/s12291-020-00935-0

https://savvysciencepublisher.com/jms/index.php/gjrc/article/view/879/842

https://www.ncbi.nlm.nih.gov/pmc/articles/PMC9707029/

Finally, it is important to note that the use of other non-standard protocols in COVID has been useful in successful treatments. For instance the use of Erythropoietin.

https://www.ncbi.nlm.nih.gov/pmc/articles/PMC7275159/

This may in part, be attributed to the avoidance of extremely aggressive therapies that may actually complicate recovery.

Round 2

Reviewer 1 Report

Translator      

no further comments

Author Response

Response to reviewer 1

We appreciate your valuable feedback. We agree with the reviewer. We review the modified narrative style, eliminating numbering and bullet points in the introduction, results and discussion.